# Unveiling Multistability in Urban Traffic Through Percolation Theory and Network Analysis

**DOI:** 10.3390/e27070668

**Published:** 2025-06-22

**Authors:** Rui Chen, Jiazhen Liu, Yong Li, Yuming Lin

**Affiliations:** 1Department of Electronic Engineering, Tsinghua University, Beijing 100086, China; akashicr@163.com (R.C.); liyong07@tsinghua.edu.cn (Y.L.); 2Department of Urban Planning and Design, School of Architecture, Tsinghua University, Beijing 100086, China; 3Technology Innovation Center for Smart Human Settlements and Spatial Planning & Governance, Ministry of Natural Resources, Beijing 100812, China

**Keywords:** urban traffic, percolation theory, network analysis, phase transition, multistability

## Abstract

Traffic congestion poses a persistent challenge for modern cities, yet the complex behavior of urban road networks—particularly multistability in traffic flow—remains poorly understood. To address this gap, we analyzed a high-resolution traffic dataset from four Chinese cities over 20 working days (5-min intervals), applying percolation theory to characterize system performance via congestion rate (*f*) and the size of the largest functional cluster (*G*). Our analysis revealed clear bimodal and multimodal distributions of *G* versus *f* across different periods, ruling out random failure models and confirming the presence of multistability. Leveraging data-driven clustering and classification techniques, we demonstrated that road segments with high betweenness centrality are disproportionately likely to become congested, and that the top 1% most topologically important roads accurately predict both stable state types and the joint behavior of *G* and *f*. These findings offer the first large-scale empirical evidence of multistability in urban traffic, laying a quantitative foundation for forecasting phase transitions in congestion and informing more effective traffic management strategies.

## 1. Introduction

The urban road network is a prime example of a complex network in which road segments and intersections intricately link together, leading to complex topological arrangements [1,2]. In a complex network with multi-scale characteristics and dynamic fluctuations of traffic flow, traffic congestion is not solely a phenomenon at individual road segments. Still, it manifests interconnection on the whole network, showing cascading effects and emergent phenomena [3]. In 2020, the economic losses caused by traffic congestion in Chinese cities reached as high as 250 billion RMB, accounting for 5–8% of the GDP [4]. Therefore, traffic congestion is an unavoidable problem for urban managers and planners, and comprehending the congestion mechanisms from a complex network perspective is crucial for solving the problem.

Despite adequate research on the scaling law [5,6,7], cascading effects [3,8,9], and phase transitions [10,11,12] of traffic congestion, there are still some important characteristics of complex networks that have not been well investigated in road networks yet. The multistability phenomenon is one of these, which has been observed in diverse complex systems encompassing ecological [13,14,15,16], climatic [17], economic [18], biological [19], and physical [20] domains. Multistability refers to the presence of several stable equilibrium states in a complex network, which forms several basins of attraction that are robust to small perturbations [16,17]. Singular abrupt perturbation may lead to a critical transition from a desired stable equilibrium to an undesired one in which the system function is impaired or lost [21]. Researchers have found that the size, shape, and depth of multiple stable equilibria can be used as global stability or resilience measures, pinpoint critical transitions, and offer the potential for early detection [22,23,24,25]. Although multistability is basic and critical for complex networks, there is limited research on multistability in urban traffic networks. Zeng et al. [26] first reported three meta-stable regimes in traffic systems, but limited to the megacities of Beijing and Shanghai.

On the other hand, percolation theory is one of the most useful tools in characterizing the phase transition between stable equilibrium. Percolation describes the geometric phase transition of a network when nodes or links are removed, in which significantly larger components will suddenly collapse into small disconnected clusters if the removal probability is above the tipping point [10,12]. The percolation framework has demonstrated significant advantages in urban traffic congestion, since congestion can be naturally viewed as segment removal from a road network, and the whole road network would collapse into small disconnected clusters as congestion gets broader. With percolation theory, the largest connected component size (*G*) can represent the functional state of traffic [12,27], and the analysis of the congestion rates (*f*) can provide a precise assessment of traffic states and resilience [12,28]. Percolation theory offers an intuitive physical-based model that reflects the topological influence of road networks and aligns seamlessly with urban characteristics such as scaling laws [28]. However, the percolation theory is limited when investigating multistability [26,29]. As a physical model based on grid networks, it may not fully capture the complexity and diversity of real urban road networks. The binary distinction of percolation and non-percolation around the tipping point may hinder the understanding of multistability. Furthermore, the stochastic processes used may over-simplify the intricacy of traffic.

The data-driven method has significant potential to solve the gap between theoretical percolation models and the empirical world. With the abundant data collected, it is possible to verify multistability directly from observations, avoiding binary distinction on percolation [26]. Furthermore, network analysis methods can provide advanced descriptions of the topological features of road networks, like the betweenness centrality [30], as replacements for the grid network used in percolation. Besides, the empirical observations may differ from the stochastic processes assumed and help to identify more critical influential factors. Finally, data-driven machine learning technology can also provide a strong ability to predict and provide practical suggestions on congestion [31,32].

In this study, our investigation delves into multistability within urban traffic networks through the percolation theory with the data-driven approach. The analysis used a high-resolution dataset of real urban traffic from 4 cities over 20 working days, capturing information at 5-min intervals. We employed the percolation theory, utilizing congestion rates (*f*) and the largest connected component size (*G*) to assess the functional status of urban traffic. Results show distinctive bimodal or multimodal distributions of *G* condition to *f* across times. This observed deviation from stochastic removal corroborates the existence of multistability. Furthermore, our comparative analysis across cities establishes a correlation between multistability and the road network topology. Employing data-driven techniques, we uncovered the nexus between topological significance and the probability of congestion for specific road segments. Based on multiple stable state types differentiated by the Density-Based Spatial Clustering of Applications with Noise (DBSCAN) clustering algorithm, it is observed that roads with high betweenness centrality had a higher probability of congestion. By leveraging the eXtreme Gradient Boosting (XGBoost) classification algorithm, we further verified that the top 1% of topologically important road segments effectively reflect both global stable types and provided accurate estimations of *G* and *f*. In summary, our study contributes a comprehensive identification and exploration of multistability within traffic networks based on a large-scale dataset, providing the basis for forecasting urban traffic situations and optimizing traffic management strategies.

## 2. Materials and Methods

### 2.1. Data Source

Our dataset is provided by the Gaode platform in Beijing, China. It contains over 20 working days from 4 Chinese cities, namely Nanjing, Jinan, Chengdu, and Wuhan. The dataset comprises road congestion index (0 for congested, and 1 for normal traffic flow) data at a time resolution of 5 min. The road topology data for these cities is sourced from OpenStreetMap [33], where continuous road segments are represented as nodes and road connectivity serves as the edges. As a road segment transitions from free-flowing to congested, we follow the percolation theory to remove the associated node from the network, thereby simulating the congestion propagation and investigating the multistability within the road network as it undergoes disintegration. Table 1 shows the basic information of the road network used in this study.

This high-resolution dataset covering multiple cities supports observing detailed variations in overall urban traffic congestion, which is crucial in understanding the spatiotemporal patterns of traffic phase transitions.

### 2.2. Method

Based on the percolation theory [26], we use *G*, denoting the number of road segments within the largest functional road subgraph, to reflect the functional state of the urban traffic network. Furthermore, we take the ratio of the number of congested roads to the total number of roads as the congestion rate *f*. The *G* and *f* are normalized to the range 0 to 1, representing the empirical observation of the road network, and can be obtained under each time snap of each day. Then we systematically analyze the connectivity of the urban road network by progressively removing road segments according to their congestion status. At each time snapshot, we represent the traffic network as a graph G=(V,E), where *V* is the set of road intersections and *E* is the set of road segments. For each observed congestion rate *f*, we remove a fraction *f* of edges (road segments) corresponding to the most congested roads, and compute the size of the largest connected component G(f) as a measure of network integrity. This process is repeated for all time points, resulting in a time series of G(f). To assess whether the observed *G*-*f* relationship is non-trivial, we conduct 1000 Monte Carlo random removal simulations for each *f*, randomly selecting the same fraction of edges and recording the resulting largest component size $G_{r}$. The comparison between empirical and randomized results allows us to identify signatures of non-random, possibly multi-stable, network behaviors.

It is also important to feature and group the observed conditions into different states, which can be achieved using clustering algorithms. The key idea is that observations belonging to the same state would cluster together so that each state forms a distinct cluster in feature space [34]. Here, we choose the Density-Based Spatial Clustering of Applications with Noise (DBSCAN) [35]. The DBSCAN regarded two points as neighbors if their distance is smaller than the parameter radius ϵ. Points with many neighbors will be clustered, and outliers with too few neighbors will be left as noise, where the threshold is set by a parameter minPts. In this study, we follow the previous literature to determine the ϵ and minPts through the highest minimum silhouette value [36], which will give us multiple stable states that stand as different attract basins.

Specifically, for every time snapshot, we constructed a two-dimensional feature vector consisting of the normalized size of the largest connected component (*G*) and the corresponding congestion rate (*f*). Prior to clustering, these features were standardized to zero mean and unit variance to ensure consistent distance measurement. To robustly determine the DBSCAN hyperparameters—specifically the neighborhood radius (ϵ) and the minimum number of neighbors (minPts)—we performed a grid search across a range of candidate values and selected the parameter combination that maximized the average silhouette score, thereby ensuring optimal cluster separation and validity. Using the optimal parameters, DBSCAN was applied to the standardized (G,f) data for each city, assigning each observation to a cluster or labeling it as noise. In this context, clusters correspond to high-density regions in feature space and are interpreted as distinct stable states of the traffic network, whereas noise points are regarded as transitional or unstable periods. The resulting cluster assignments allow us to visualize and quantify the number, range, and transition patterns of stable states in each city, and to further analyze their association with underlying road network topologies.

Network analysis based on graph theory provides effective indicators for an in-depth understanding of multistability; among them, the betweenness centrality measure is suitable for our purpose. Betweenness centrality *B* of a node *v* measures the acting as a bridge along the shortest path between two other nodes, given by$$B\left(v\right)=\sum_{s\neq t\neq v}\frac{\sigma_{st}\left(v\right)}{\sigma_{st}},$$
where $\sigma_{st}$ is the total number of shortest paths from node *s* to node *t*, and $\sigma_{st}\left(v\right)$ is the number of those paths that pass through node *v*. In our scenario, road segments act as nodes and bridge traffic flow to other places, making betweenness an excellent indicator to represent the intrinsic attributes and establish connections to the global states.

Given the intricate nature of multistability, our inquiry extends to utilizing the congestion status of select roads to forecast the overall stable states. This stems from recognizing that exhaustive monitoring of the entire road network is resource-intensive and time-consuming. By extrapolating the broader situation from localized assessments, we can efficiently identify critical segments and bottlenecks warranting management intervention, as well as optimizing resource allocation. Here, we choose the eXtreme Gradient Boosting algorithm (XGBoost) [37] for the prediction task, which provides a variant of the Gradient-Boosted Decision Tree method that iteratively combines the predictions of multiple decision trees to enhance predictive accuracy.

Specifically, critical road segments were first identified based on their betweenness centrality; we systematically selected the top 0.05%, 0.1%, 0.5%, and 1% of road segments with the highest betweenness values in each city. For each time snapshot, the binary congestion status of these critical roads was used as the feature vector. Three prediction tasks were formulated: (i) classifying the global stable state label as determined by the DBSCAN clustering (i.e., identifying which attractor basin the current traffic network belonged to), (ii) regressing the global congestion rate *f*, and (iii) regressing the normalized size of the largest functional road cluster *G*. For each task, we constructed feature matrices using the historical congestion status of the selected road segments, and the corresponding target variables (cluster label, *f*, or *G*). The entire dataset was randomly split into training and testing sets with a 7:3 ratio. XGBoost hyperparameters (e.g., tree depth, number of estimators, learning rate) were optimized using cross-validation on the training set. The classification task was evaluated using accuracy, area under the curve (AUC), and F1-score, while the regression tasks were assessed by mean squared error (MSE), mean absolute error (MAE), and the coefficient of determination ($R^{2}$). To evaluate the effect of feature sparsity, all four selection thresholds for key road segments were tested, and model performance was compared accordingly.

## 3. Results

### 3.1. Non-Uniqueness of Percolation States

The traffic system of Nanjing City will be used as an example without losing generality. Nanjing is an important city in eastern China, with a population of 9 million people and an extensive road network serving over 200 km^2^. The city is located along the Yangtze River and is characterized by multiple cross-river bridges and tunnels that connect the two banks of the city. During peak hours, the dense traffic flow across these river crossings often causes severe traffic congestion.

Through percolation analysis, it can be observed that the largest functional cluster *G* decreases as the congestion rate *f* increases (Figure 1). Diverging from conventional binary percolation analysis, the identical *f* of a real urban traffic network may correspond to multiple distinct ranges of the *G*.

Taking the example of two different observations in Figure 1A, it is shown that the *G* exhibits two distinct performance levels with roughly equal congestion rate f=0.04. Specifically, when the road connecting two subregions is functioning normally, the urban traffic can form a large functional cluster (Figure 1B), representing a high-performance state of the traffic network. However, when the road connecting two subregions fails, the largest functional cluster collapses into smaller clusters (Figure 1C), representing a low-performance state of the traffic network. This behavior can be regarded as a traffic phase transition induced by critical road failures.

Further analysis shows a clearer multistability phenomenon. As shown in Figure 1E, for lower congestion rates (f=0.02), urban traffic exhibits a bimodal distribution of *G*, with a higher probability of being in the high-performance state that ranges from 0.8 to 1.0. As *f* increases to 0.03 (Figure 1G), more roads become congested, increasing the probability of being in the low-performance state that ranges from 0.6 to 0.8. The traffic system dynamically oscillates between these two states (Figure 1D,F) but without an intermediate state. Furthermore, when the *f* exceeds a certain threshold of around 0.04, the traffic system abruptly collapses and enters a variable intermediate state where *G* spreads widely (Figure 1A). When *f* > 0.06, the traffic system becomes exceedingly fragile, with the *G* staying nearly 0, which implies that, unlike traditional percolation processes, the phase transition does not occur instantaneously but exists as a switch among multiple stable states.

### 3.2. Multimodal Distribution of Network States Indicators

To further explore the mechanisms underlying the multistability of traffic systems, we conducted a comparative analysis between random simulations and empirical data. As mentioned, we employed a node removal with 1000 instances at various *f*, and the size distribution of $G_{r}$ was calculated and contrasted with empirical data.

To better understand whether the observed multistability results from intrinsic network properties or dynamic processes, we compare the empirical percolation curves with three simulation baselines in Figure 2: (i) random removal of road segments, (ii) removal based on static betweenness centrality, and (iii) removal based on weighted betweenness centrality. The random removal serves as a structure-free baseline, while the betweenness-based strategies simulate structurally informed degradations using topological importance rankings. Surprisingly, the empirical curves consistently show faster and more irregular network breakdowns than the simulations. In other words, even when roads are sequentially removed by descending betweenness centrality—a strategy often associated with high vulnerability—the real-world congestion process still leads to earlier fragmentation and greater volatility in network connectivity. This indicates that the observed multistability cannot be attributed solely to static network properties. Instead, it reflects the influence of dynamic traffic interactions, local congestion propagation, and spatiotemporal heterogeneity in road usage patterns.

### 3.3. City-Specific Multistable States and Network Structures

To further investigate the multiple stable states and reduce the disturbance from noise, we employed the DBSCAN mentioned earlier to identify the potential stable states within the urban traffic systems. We assumed that the observation of stable states should form high-density clusters while transitional periods exhibit relatively sparse noise. Figure 3 reveals the presence of varying numbers and ranges of stable states within different cities, suggesting a correlation with the unique road network topology of each locale. Besides their statistical significance obtained via DBSCAN clustering, these functional states carry explicit physical and practical implications closely linked to urban network performance. For instance, a road network that exhibits strong connectivity, high resilience, and efficient flow, with minimal or localized congestion corresponds to high-performance states, while global collapse states suggest severe congestion at critical network points significantly disrupts connectivity. Moreover, there is an intermediate transitional state, which implies moderate congestion in regions containing distinct sub-networks or critical bridge-like segments.

As shown in Figure 3A, the traffic states in Nanjing can be categorized into four distinct clusters: two distinct stable states, characterized by high and low performance of *G* with *f* below 0.05, respectively rendered in blue and orange. Subsequently, it abruptly transitions into a variable intermediate state in green before rapidly deteriorating into a global collapse state in red. On the other hand, the case of Jinan and Chengdu shows only two primary states, global functional and global collapse, respectively. When the *f* is below 0.06, the Chengdu network generally operates at a global functional state, shown in blue (Figure 3C). When *f* is in the range of 0.06 to 0.08, the system is characterized by a lagging and variable behavior but with a clear boundary and no intermediate state. As *f* increases, the system stabilizes into the global collapse state. Jinan shares similarities with Chengdu but with a lower *f* threshold, along with a highly sparse intermediate variable state (Figure 3B) between the global functional and global collapse states. Meanwhile, Wuhan’s traffic phase transition is unique, manifesting as many as five clusters, identifying multiple stable states (Figure 3D). Given that Wuhan is dissected by the Yangtze River, Han River, and numerous water bodies, such distinct patterns likely stem from the city’s unique network topology.

Based on the topology and degree distribution of multiple cities, we have observed a significant correlation between multistability patterns in different cities and their respective topological features. The presence of an intermediate state is contingent upon the road network being composed of several major components; otherwise, there is no intermediate state. Furthermore, different road segments play significantly varying roles in the traffic phase transition. Observing the road network structure of Nanjing (Figure 3A), the failure of the road connecting two banks of cities will lead to an abrupt transition from the high-performance state to the low-performance state. Similarly, in Wuhan (Figure 3D), multiple sub-regions are connected by relatively few road or bridge segments. Once these critical segments experience congestion, the network immediately transitions to another state, resulting in multiple stable states. The road network of Chengdu exhibits a concentric ring structure resembling a spider’s web (Figure 3C). Similarly, Jinan consists of a cohesive grid-like structure (Figure 3B). Such structures would not be easily affected by a single segment, exhibiting only two primary states. Our result shows that the multistability observed in different cities has a strong relationship with their respective topological features, which requires further microscopic analysis.

### 3.4. Multistable States and Congested Road Segments

To quantitatively evaluate which network centrality measure best captures congestion vulnerability under different system states, we compare three commonly used metrics: betweenness centrality, degree centrality, and eigenvector centrality. For each metric, we divide road segments into four levels based on their percentile ranks and examine their congestion probabilities across two functional states (middle and low), using empirical data from Wuhan. The results are shown in Figure 4.

The comparison reveals a clear distinction: under the low functional state, road segments with the highest betweenness centrality (level 4) exhibit the highest probability of congestion, suggesting that these segments act as critical flow bridges whose failure contributes directly to system-wide breakdown. In contrast, segments with high degree or eigenvector centrality show the opposite trend: they are less likely to be congested under low-performance conditions, likely due to their structural redundancy or multiple alternative routing paths.

In middle functional states, the differences across centrality levels are less pronounced, but the trend for betweenness remains directionally consistent. These findings support the view that betweenness centrality is the most informative indicator of structural vulnerability in urban road networks under congestion stress. Its ability to identify congestion-prone links aligns well with the dynamics of functional state transitions observed in earlier sections, justifying its central role in our prediction and monitoring framework.

### 3.5. Predictive Power of Road Centrality in Network States

Given the close correlation between the congestion probability of high B(v) roads and the overall states, a natural inclination arises to utilize the status of these roads to forecast the global state of the network. Such an approach could further validate the significance of key segments and offer a novel avenue for traffic management and road monitoring. Therefore, we utilized congestion information of critical segments, which were selected based on betweenness centrality, to predict state labels from DBSCAN, as well as specific values for congestion rate *f* and the size of the largest functional cluster *G*.

In real-world traffic monitoring systems, the coverage of sensing devices such as loop detectors, video cameras, or GPS-based floating car data is often limited to a small subset of road segments due to installation costs, maintenance burdens, and data transmission constraints. Studies show that in many large cities, the monitored proportion of roads can be as low as 0.5% to 1% of the total network [38,39]. Therefore, our selection of the top 0.05% to 1% high-betweenness roads aligns with these practical constraints and simulates realistic sensing conditions for forecasting global traffic states.

We utilized road segments with the top 0.05%, 0.1%, 0.5%, and 1% B(v) values in each instance to perform predictions, validating the predictive influence of critical roads during traffic phase transitions. Figure 5 shows the top 0.05% to top 1% roads in the road network of Nanjing, which is only a small part of the road network and spatially concentrated. The XGBoost machine learning model is chosen to capture complex patterns and correlations within traffic data, using those The historical congestion data from the selected roads is treated as data samples. The data is partitioned into training and testing sets in a 7:3 ratio.

The results in Table 2 show that the predictions have achieved a notably high level of accuracy. Three performance metrics are employed to assess the model’s predictive capabilities, namely Accuracy, Area Under the Curve (AUC), and F1-score. Distinct performance characteristics of different selection strategies can be observed across the test and training datasets. Notably, the XGBoost model demonstrates commendable performance on both the training and test datasets, indicative of its robust generalization capacity. As an illustration, for the test set, when selecting only the top 0.05% of road segments, which includes only about 100 segments, the model achieves an accuracy of 0.8410, meaning that it can predict the global state at more than 84% accuracy. With an increasing proportion of segments, the performance gradually improves. When the top 1% of features are selected, the accuracy, AUC, and F1-score reach 0.9446, 0.9557, and 0.9405, respectively. This highlights that a subset of just 0.05% of critical road features can yield a satisfactory prediction accuracy on the global stable states, while incorporating more congestion data for critical roads significantly enhances model performance.

Subsequently, focusing on the predictive outcomes for the target variables *G* and *f*, as shown in Table 3 and Table 4, a similar trend is observed. As the proportion of segments increases, the Mean Squared Error (MSE) and Mean Absolute Error (MAE) on the test set gradually decrease, while the coefficient of determination ($R^{2}$) gradually increases. When selecting the top 1% of segments, the model achieves its optimal performance. For the prediction of *G*, the MSE is 0.0158, MAE is 0.0740, and $R^{2}$ is 0.8698. In the case of predicting *f*, the MSE is 0.0001, MAE is 0.0059, and $R^{2}$ is 0.9546. This underscores that even for specific predictions of the explicit global state *G* and *f*, a smaller subset of critical road segments contains enough information to yield precise predictive results. We may conclude that the status of critical road segments within the network predominantly determines the boundaries of distinct stable states, which shed light on traffic management.

## 4. Discussion

Our research demonstrates the existence of multistability within urban traffic systems, along with the influence of road network topology on traffic phase transitions. Firstly, we observe that, in contrast to traditional percolation simulations, real urban traffic networks may correspond to multiple distinct largest functional cluster sizes *G* under the same congestion rate *f*. The key indicator *G* follows distinct, often bimodal or multimodal, probability distributions, which exhibit greater stability compared to outcomes resulting from random removal. Then, we identify several distinct clusters based on the *G*-*f* graph using DBSCAN clustering method, indicating the presence of multiple stable states within urban traffic networks in different cities. Results from multiple cities illustrate the varying numbers of states with different *G*-*f* ranges, closely related to the cities’ road network topology. Further analysis finds that road segments with high betweenness centrality behave significantly differently under different stable states, inspiring us to utilize the top 1% betweenness road segments to predict the state label and the value of *G* and *f*. This underscores that the topological characteristics of a city’s road network, along with the states of these critical road segments, significantly determine the number of distinct stable states and the precise boundaries of these states.

The results of this study hold significant implications. We empirically confirm the existence of multistability in traffic systems across diverse cities, establishing a clear link between these states and urban road network topology, particularly betweenness centrality. This provides a crucial foundation for subsequent percolation theory research. Secondly, our approach extensively employs empirical data and machine learning, elucidating the close connections among various novel methodologies and highlighting the vital role of interdisciplinary research. Lastly, our study holds importance for urban resilience analysis. By extending beyond the binary phase transition point implied by traditional percolation theory, the multistability framework is better suited to describe the potentially more intricate processes of resistance, adaptation, and recovery, thereby expanding the definition and boundaries of traffic resilience.

More importantly, the findings of this study offer practical implications for urban and transportation planning. First, we demonstrate that only a very small fraction of road segments—specifically, those with the highest betweenness centrality—are required to accurately predict the global traffic state. This highlights the feasibility of low-cost, targeted sensor deployment and selective monitoring strategies that focus on structurally critical segments to achieve system-wide awareness. Second, the observed multistability patterns are strongly influenced by the topology of urban road networks. Cities with highly modular or bridge-like structures (e.g., Wuhan, Nanjing) exhibit more distinct stable states and abrupt transitions, while grid-like or concentric structures (e.g., Jinan, Chengdu) are more resilient. This suggests that planners should incorporate topological resilience considerations into road network design, especially when developing new infrastructure or retrofitting existing networks. Third, the ability to predict system states from partial observations offers a foundation for proactive traffic governance. By identifying critical segments and forecasting transitions in real time, planners can implement early warning systems and targeted interventions to avoid global breakdowns. This approach aligns with the broader goals of smart city development and adaptive traffic control.

Our study has certain limitations. Firstly, our traffic dataset is primarily concentrated within China, and future research could extend this study to encompass a broader range of cities worldwide. Secondly, our predictions based on the XGBoost solely relied on congestion data from critical roads, which did not fully leverage topological information. In future research, more complex models considering the topology structure, such as Graph Neural Networks (GNN), could be integrated. Thirdly, our study lacks a unified model across cities due to the inherent uniqueness of each urban environment. Finally, our research did not employ physical modeling, which may hinder the deep understanding of the transition process. Future investigations could incorporate centrality modeling into percolation processes to physically reproduce empirical data and validate the findings.

Based on the aforementioned research findings, this study holds broad applications. Firstly, it offers profound insights into urban traffic planning and management by unveiling the presence of multistability within traffic networks. This understanding contributes to more effective congestion management and enhances urban sustainability. Furthermore, the predictive results highlight the decisive role of critical bottlenecks within traffic networks in determining the overall operational states of urban traffic systems. Leveraging high-betweenness road segments for traffic prediction can significantly enhance forecasting accuracy, with potential applications in real-time traffic management and navigation systems. Additionally, this study has significant potential in disaster warning and extreme-condition evacuation operations. By gaining deeper insights into the multistability characteristics of urban traffic networks, we can more effectively plan evacuation routes, optimize resource allocation, and provide flexible warning systems, bolstering urban resilience, mitigating potential risks, and minimizing losses. Consequently, this research not only contributes to the improvement of everyday traffic management but also provides robust support for urban safety and sustainability, particularly in emergency scenarios.

## Figures and Tables

**Figure 1 entropy-27-00668-f001:**
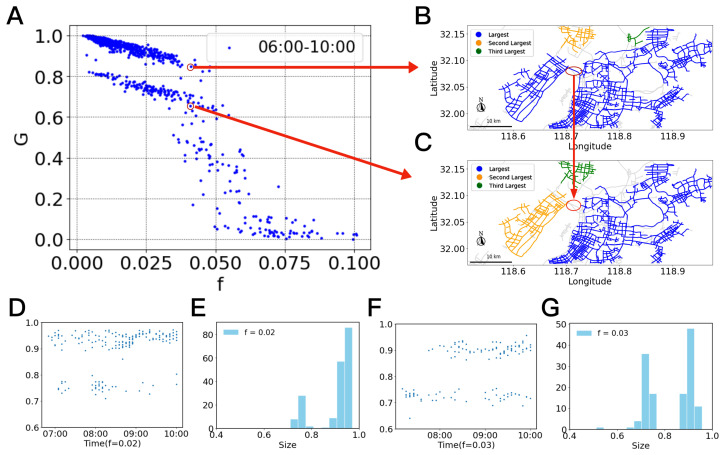
Example of urban traffic phase transition based on real traffic data in Nanjing (all weekdays from 5 March to 5 April 2022). (**A**) The *G*-*f* plot for rush hours 06:00 to 10:00 AM on all working days. (**B**) The road network in a high performance state when f=0.04 (blue represents the largest functional cluster, orange represents the second largest, and green represents the third largest). (**C**) The road network topology in a low functional state when f=0.04. (**D**) The relative *G* for a given f=0.02, evolves with time between essentially two states (5 March 2022). (**E**) The distribution of *G* when f=0.02. The bimodal phenomenon is seen. (**F**) The relative *G* for a given f=0.03, evolves with time between essentially two states (5 March 2022). (**G**) The distribution of *G* when f=0.03. The bimodal phenomenon is seen.

**Figure 2 entropy-27-00668-f002:**
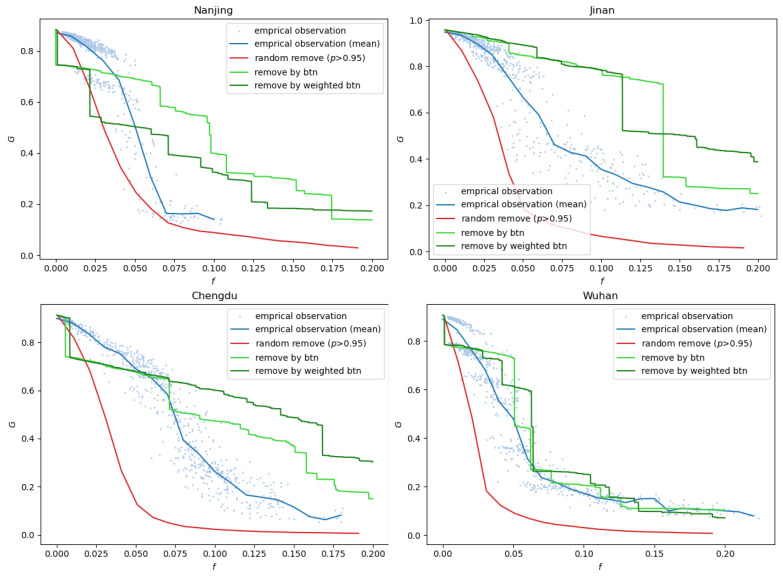
Percolation curves of four cities comparing empirical observations and simulation-based removals. Blue points and lines represent real-world congestion patterns (*G* vs. *f*); red lines show random removal results; green and dark green lines correspond to removals based on static and weighted betweenness centrality.

**Figure 3 entropy-27-00668-f003:**
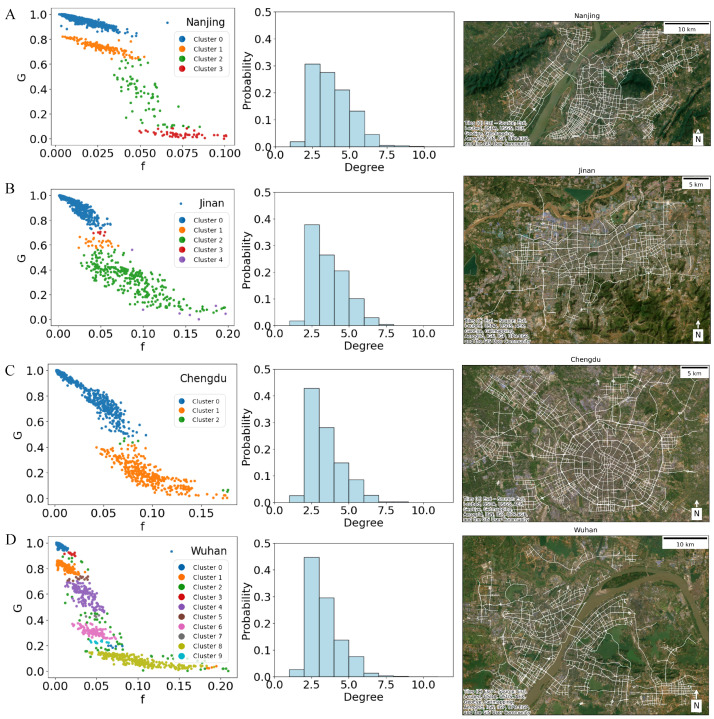
Examples of city-specific multistable states, the corresponding degree distributions, and road networks. The *G*-*f* plot shows rush hours from 06:00 to 10:00 AM on all of the working days, rendered by clustering results from DBSCAN in (**A**) Nanjing, (**B**) Jinan, (**C**) Chengdu, and (**D**) Wuhan.

**Figure 4 entropy-27-00668-f004:**
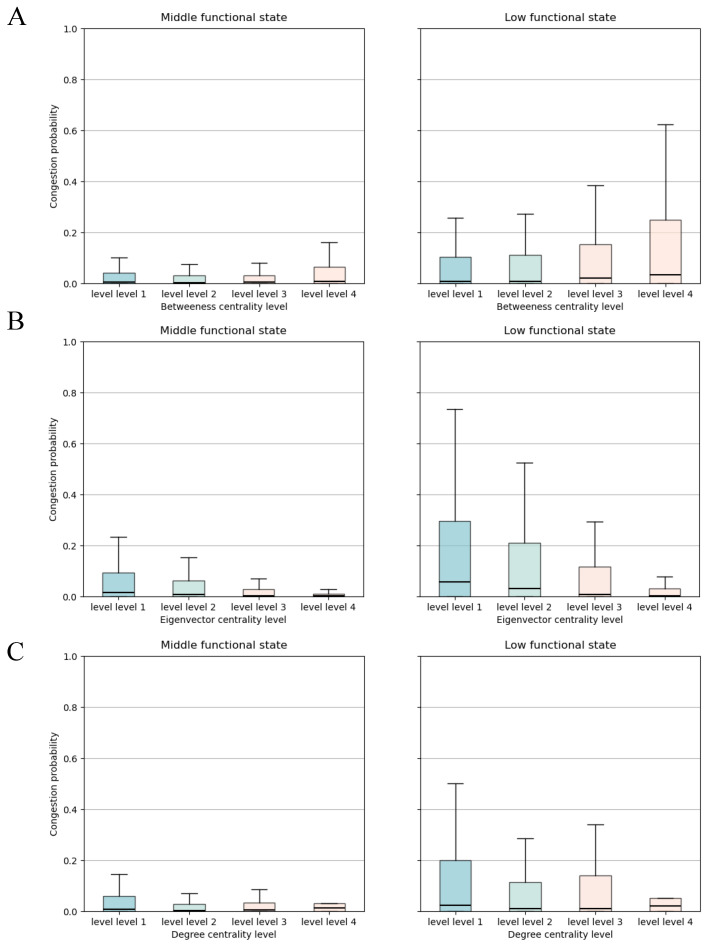
Comparison of congestion probability across different network centrality levels in Wuhan, grouped by functional traffic states. (**A**) Betweenness centrality: Under the low functional state, roads with the highest betweenness centrality (level 4) show the highest congestion probability, indicating their critical role in traffic flow breakdown. (**B**) Eigenvector centrality: Roads with higher eigenvector centrality tend to exhibit lower congestion probability, especially in the low functional state, suggesting potential redundancy or distributed load. (**C**) Degree centrality: A similar inverse pattern is observed—high-degree roads are less likely to be congested during network collapse.

**Figure 5 entropy-27-00668-f005:**
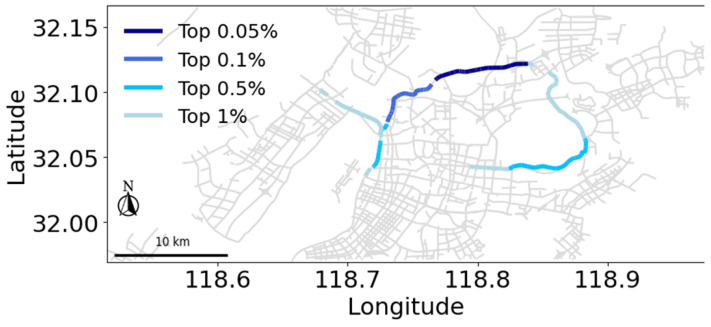
Illustration of the top 0.05% to top 1% roads in the road network of Nanjing in terms of betweenness centrality.

**Table 1 entropy-27-00668-t001:** Number of urban road sections and intersections.

City	Number of Road Segments	Number of Intersections
Nanjing	19,394	32,116
Chengdu	22,966	33,737
Jinan	11,585	17,936
Wuhan	24,305	34,555

**Table 2 entropy-27-00668-t002:** The predicted results of the training set and testing set for stable state labels.

	Test Set			Training Set		
Segments	Accuracy	AUC	F1-Score	Accuracy	AUC	F1-Score
Top 0.05%	0.8410	0.7656	0.7939	0.8624	0.7850	0.8206
Top 0.1%	0.8842	0.8220	0.8595	0.8966	0.8253	0.8809
Top 0.5%	0.9436	0.9479	0.9429	0.9753	0.9674	0.9743
Top 1%	0.9446	0.9557	0.9405	0.9876	0.9924	0.9874

**Table 3 entropy-27-00668-t003:** The predicted results of the training set and testing set for *G* in Nanjing.

	Test Set			Training Set		
Segments	MSE	MAE	$R^{2}$	MSE	MAE	$R^{2}$
Top 0.05%	0.0359	0.1471	0.7042	0.0357	0.1363	0.7098
Top 0.1%	0.0266	0.0992	0.7808	0.0115	0.0653	0.9060
Top 0.5%	0.0204	0.0864	0.8320	0.0069	0.0463	0.9441
Top 1%	0.0158	0.0740	0.8698	0.0038	0.0340	0.9688

**Table 4 entropy-27-00668-t004:** The predicted results of the training set and testing set for *f* in Nanjing.

	Test Set			Training Set		
Segments	MSE	MAE	$R^{2}$	MSE	MAE	$R^{2}$
Top 0.05%	0.0004	0.0162	0.7542	0.0005	0.0162	0.7303
Top 0.1%	0.0003	0.0121	0.8172	0.0002	0.0105	0.8733
Top 0.5%	0.0002	0.0104	0.8728	0.0002	0.0091	0.9031
Top 1%	0.0001	0.0059	0.9546	0.0001	0.0053	0.9650

## Data Availability

The original contributions presented in the study are included in the article/Appendix A, further inquiries can be directed to the corresponding author.

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
