# Peer review of "Unveiling Multistability in Urban Traffic Through Percolation Theory and Network Analysis"

_entropy, 2025, doi:10.3390/e27070668_

Round 1
Reviewer 1 Report
Comments and Suggestions for Authors
Revealing network complex behaviors for transportation systems is essential for traffic management and planning. This study investigates multi-stability in urban road networks based on percolation theory and machine learning. Four urban networks were utilized to valid the methodology and achieved findings. Overall, the paper is well-written and interesting. I have some comments for authors to consider:
(1) The methodology part could be strengthened. Although the methodology of this study is built on existing techniques, it is better to provide more details about the algorithm description related to percolation theory, stable state clustering, and stable state forecasting. This can provide more information for the readers and make the achieved findings convincible.
(2) In the experimental part, why are the both empirical and simulation data to reveal the multi-stability in the real-word network. As seen from Figure 2, the two datasets reflect different system characteristics.
(3) In Section 3.5, the road segments with the top 0.05%-1% were selected to develop the forecasting models. As can be observed in Tables 2-4, the classification and forecasting accuracy promote with the increase of number of selected road segments. Is there any proper threshold associated with the number of selected road segments for developing the models?
Author Response
We wish to thank the Reviewer for her/his valuable comments to the manuscript:
- Comments 1: The methodology part could be strengthened. Although the methodology of this study is built on existing techniques, it is better to provide more details about the algorithm description related to percolation theory, stable state clustering, and stable state forecasting.This can provide more information for the readers and make the achieved findings convincible.
- Answer 1:
We sincerely thank the reviewer for the valuable suggestion regarding the methodology section, especially the emphasis on providing more algorithmic details to enhance the transparency of our study.
In response, we have substantially revised and expanded the Methods section to offer a more comprehensive and detailed account of the percolation analysis, stable state clustering, and state prediction procedures. Specifically, the following improvements have been made:
- Percolation Analysis:
We have clarified the algorithmic steps for constructing the urban road network, calculating the empirical and random percolation curves, and performing targeted removals based on betweenness centrality. The entire workflow is now summarized as a structured pseudocode (see Algorithm 1). We also add a paragraph to illustrate the algorithm details in the manuscript Page 3, lines 110-121.
- Stable State Clustering with DBSCAN:
We have expanded our description of the DBSCAN clustering process, explicitly detailing the input feature construction, preprocessing (feature standardization), parameter optimization (using a grid search and silhouette score maximization), and the physical interpretation of cluster assignments. The entire workflow is now summarized as a structured pseudocode (see Algorithm 2). We also add a paragraph to illustrate these details in the manuscript Page 4, lines 132-146.
- Stable State Forecasting with XGBoost:
We now offer a detailed account of how key road segments are selected via betweenness centrality, how feature vectors are constructed, and how state labels (from DBSCAN) and global indicators (f and G) are predicted using the XGBoost model. The entire workflow is now summarized as a structured pseudocode (see Algorithm 3). We revised the manuscript in Page 4-5 to specify the training/testing data split, hyperparameter tuning strategy, and evaluation metrics employed, lines 164-180. The code of XGBoost algorithm is adopted from Ref [37].
The relevant sections and algorithms are highlight in red in the revised manuscript for your convenience. We believe these additions address the reviewer’s concern and significantly improve the methodological clarity and rigor of our work. We hope the revised methodology now meets the expectations of the reviewers and the journal.
- Comments 2: In the experimental part, why are the both empirical and simulation data to reveal the multi-stability in the real-word network. As seen from Figure 2, the two datasets reflect different system characteristics.
- Answer 2:
We thank the reviewer for this insightful question regarding the purpose and interpretation of combining both empirical and simulation-based data in our analysis of multistability.
We apologize for the ambiguity of the original Fig. 2. To better illustrate our findings, we redraw the Fig. 2.
This new figure is designed to serve as a comparative framework that highlights the unique characteristics of real-world traffic systems in contrast to synthetic percolation scenarios. Specifically:
- The random removal curve (red) represents that congested roads are removed in a purely stochastic manner, showing a sharp decrease in connectivity.
- The betweenness-based removal curves (green and dark green) simulate targeted attacks on the most structurally critical road segments, assuming prior global knowledge of the network topology.
- The empirical observation curve (blue), however, sits between these two extremes, suggesting that real traffic dynamics present faster and more abrupt collapses than simulations based on fixed centrality rankings but slower collapses than random removal.
This divergence highlights a key insight: the observed multistability and collapse patterns in real-world urban traffic are not merely consequences of network topology or statistical randomness. Instead, they emerge from dynamic, decentralized congestion propagation mechanisms, behavioral feedbacks, and real-time network stress.
We have revised the text near Figure 2 (page 6, lines 218-231) to better explain this distinction and the rationale behind combining empirical and simulated percolation curves for comparative analysis.
- Comments 3: In Section 3.5, the road segments with the top 0.05%-1% were selected to develop the forecasting models. As can be observed in Tables 2-4, the classification and forecasting accuracy promote with the increase of number of selected road segments. Is there any proper threshold associated with the number of selected road segments for developing the models?
- Answer 3:
We thank the reviewer for raising this valuable point. To better identify an appropriate threshold for the number of selected road segments, we have conducted additional analysis illustrating how predictive accuracy varies with the proportion of selected segments (p). We find that the threshold is around p~0.1%-0.5%. When p<0.1%, the accuracy of the model sharply increase with p. In contrast, when p~0.1%-0.5%, the accuracy increase slowly with p. When p>0.5%, the accuracy of the prediction will gradually saturated. We show the relevant plot Fig. S1 in the Supplementary Materials.
This finding holds practical implications for urban planning and traffic management. In realistic scenarios, the cost constraints typically limit sensor coverage to at most 1% of road segments. Our analysis demonstrates that reliable and accurate state predictions can be achieved using only 0.1%–0.5% of the critical road segments, which is significantly lower than this practical upper limit. We have made relevant discussions in the revised manuscript discussion section (lines 367-382) and added Fig. S1 to the supplementary materials.
Reviewer 2 Report
Comments and Suggestions for Authors
The article " Unveiling Multistability in Urban Traffic through Percolation Theory and Network Analysis" delivers some interesting insights. For this round, I would recommend a major revision for improving presentation and results. Here are some comments on the aspects of urban planning and spatial data science:
- It would be great to provide readers with maps on the overall study area and a display of road network if possible. With these maps, we may present information from Fig 3. more clear.
- Map representation - north arrow, scale bar, legend and title are always needed (for instance Fig 1. B and C). For Fig 1. B and C, it is hard for readers to understand the meaning of color of different roads. The same also applies to Fig 3. A,B,C D, I understand that different color means different clusters, while we still need legends for these groups.
- I am not confident in seeing the findings (differences among groups) from Fig 4 A and B. It would be great if other network feature index could be added and compared. I notice that you have mentioned that 'betweenness centrality' is the most appropriate, while it would be great if solid evidence could be provided with quantitative comparison.
- Line 233: " several categories of functional states based on the results of DBSCAN", is there any physical or practical meaning of functional states other than the data science aspect using clustering? It would be great if the authors can explore more on this.
- Line 258: why road segments with the top 0.05%, 0.1%, 0.5%, and 1% B(v) values are important? does these roads have specific and special meaning in urban system? What about the rest 99%? Why the top 1% has importance over the rest 99%?
- Results and discussion - given the nature of this article with an application, I would also recommend some more content for planning. How planner could benefit from the findings and results of this article? Apart from good model performance, is there any concrete planning advice for transport and urban system given the current results?
can be improved but not necessary
Author Response
We wish to thank the Referee for raising very important questions regarding the manuscript.
Comment 1: It would be great to provide readers with maps on the overall study area and a display of road network if possible. With these maps, we may present information from Fig 3. more clear.
Answer 1:
We thank the reviewer for this thoughtful suggestion. In response, we have revised Fig. 3 that provides an overview map of the study areas, with the corresponding road networks for each city. These maps help contextualize the differences in topological features and make the interpretation of Figure 3 more intuitive. We believe this addition enhances the spatial clarity and accessibility of our findings.
Comment 2: Map representation - north arrow, scale bar, legend and title are always needed (for instance Fig 1. B and C). For Fig 1. B and C, it is hard for readers to understand the meaning of color of different roads. The same also applies to Fig 3. A,B,C D, I understand that different color means different clusters, while we still need legends for these groups.
Answer 2:
We appreciate the reviewer’s careful attention to the clarity of map representations. In response, we have revised the relevant figures (Fig. 1, Fig. 3, and Fig.5) to include essential cartographic elements such as north arrows, scale bars, and legends. We have also added explicit legends to clarify the color schemes used for different road segments and cluster assignments. These improvements aim to enhance visual interpretability and reader comprehension.
Comment 3: I am not confident in seeing the findings (differences among groups) from Fig 4 A and B. It would be great if other network feature index could be added and compared. I notice that you have mentioned that 'betweenness centrality' is the most appropriate, while it would be great if solid evidence could be provided with quantitative comparison.
Answer 3:
We thank the reviewer for their thoughtful suggestion. In response, we have extended our analysis to include two additional network centrality measures—degree centrality and eigenvector centrality—in comparison with betweenness centrality across different functional states.
The results, shown in the updated Figure 4 and Supplementary Figure S2-S38, reveal a clear and consistent pattern: under the low functional state, road segments with higher betweenness centrality exhibit the highest congestion probability, whereas segments with higher degree or eigenvector centrality tend to be less congested. This indicates that roads serving as critical connectors (i.e., those with high betweenness) are more functionally vulnerable under system stress, while highly connected nodes (high-degree) or influential nodes (high eigenvector centrality) may benefit from alternative routing options or network redundancy.
We also conduct statistical tests for different levels in the same functional states and across different functional states. The test results indicate statisitical significance (see Table S1-2). These findings offer quantitative support for our choice of betweenness centrality as the most appropriate indicator for understanding congestion dynamics and predicting global state transitions. We have rewritten Section 3.4 and included the supporting figures in the revised manuscript and Supplementary Materials.
Comment 4: Line 233: " several categories of functional states based on the results of DBSCAN", is there any physical or practical meaning of functional states other than the data science aspect using clustering? It would be great if the authors can explore more on this.
Answer 4:
We thank the reviewer for the insightful comment on the practical meaning of the identified functional states. Indeed, besides their statistical significance obtained via DBSCAN clustering, these functional states carry explicit physical and practical implications closely linked to urban network performance. Specifically, each identified state reflects a different functional status of the urban traffic system.
High-performance (global functional) states: The road network exhibits strong connectivity, high resilience, and efficient flow, with minimal or localized congestion.
Intermediate transitional states: These reflect scenarios where specific key segments experience moderate congestion, reducing the overall network efficiency but not causing complete network breakdown. Such states typically arise in cities whose topologies contain distinct sub-networks or critical bridge-like segments.
Global collapse states: Severe congestion at critical network points significantly disrupts connectivity, leading to substantial reduction or near-total loss of functional efficiency.
As illustrated explicitly in the manuscript (see Figure 3 and related discussion), different cities exhibit different stable states and transition patterns that directly correspond to their unique network topologies. For instance:
Nanjing: The presence of bridges connecting two distinct urban regions causes abrupt transitions between high-performance and collapse states upon congestion of these critical bridges.
Wuhan: With multiple sub-regions separated by rivers, congestion in the connecting segments results in multiple distinct functional states, each representing different connectivity scenarios across sub-regions.
Chengdu and Jinan: Given their highly integrated ring or grid-like topologies, these cities typically present fewer states, generally just a global functional state and a collapse state, due to their robustness against isolated congestion events.
These physical interpretations highlight that the functional states identified through clustering represent tangible, practical states of traffic system performance. Understanding these states provides valuable insights for urban planners and traffic management agencies, facilitating targeted intervention strategies based on network topology and observed congestion patterns.
We have added and clarified this physical interpretation in the revised manuscript (Section 3.3, lines 238-246).
Comment 5: Line 258: why road segments with the top 0.05%, 0.1%, 0.5%, and 1% B(v) values are important? does these roads have specific and special meaning in urban system? What about the rest 99%? Why the top 1% has importance over the rest 99%?
We thank the reviewer for raising this valuable point. Indeed, Reviewer 1 also mention this important point: why the top 1% is more important than the others?
This is because, in realistic scenarios, the cost constraints typically limit sensor coverage to at most 1% of road segments. For example, there are totally around 30K road segments in Beijing. Among these road segments, only 300 of them are detected by sensors, that is, around 1% road segments is being detected. For the normal cities, the number of road segments being detected is much smaller than Beijing because they usually have fewer budgets.
Moreover, we have conducted additional analysis illustrating how predictive accuracy varies with the proportion of selected segments (p). We find that the threshold is around p~0.1%-0.5%. When p<0.1%, the accuracy of the model sharply increase with p. In contrast, when p~0.1%-0.5%, the accuracy increase slowly with p. When p>0.5%, the accuracy of the prediction will gradually saturated. We show the relevant plot Fig. S1 in the Supplementary Materials.
Therefore, our results holds practical implications for urban planning and traffic management.
Comment 6: Results and discussion - given the nature of this article with an application, I would also recommend some more content for planning. How planner could benefit from the findings and results of this article? Apart from good model performance, is there any concrete planning advice for transport and urban system given the current results?
Answer 6: We sincerely thank the reviewer for this important and constructive comment. In response, we have added a dedicated paragraph in the revised manuscript to elaborate on the planning implications of our findings. Beyond demonstrating good model performance, our study provides concrete insights for urban and transportation planners.
First, our results show that a very small subset of critical road segments—identified through betweenness centrality—can effectively predict the global traffic state. This has direct implications for sensor deployment and traffic monitoring, as it suggests that strategic placement on key segments is sufficient for system-wide status assessment, reducing costs and improving efficiency.
Second, we find that the number and type of stable states are closely related to the structural topology of different cities. This indicates that road network design plays a critical role in determining the resilience or fragility of traffic systems. Planners can use this information to assess the vulnerability of existing layouts and to guide the design of more robust road networks in new urban areas.
Third, our forecasting framework offers a foundation for proactive traffic management, enabling early warning and targeted intervention based on partial network observations.
These additions are now included in the revised version (see Section 4, lines 367-382).
Round 2
Reviewer 1 Report
Comments and Suggestions for Authors
I am pleased to accept this version of the paper since all my comments were properly addressed.
Author Response
Thank you for your positive comments!
Reviewer 2 Report
Comments and Suggestions for Authors
Hi authors, I thank you for your feedback on all my comments and revisions. Your revisions in figures and presentations look okay, while we may need some minor improvements before publication. Here are the reasons with examples:
(1) I appreciate you getting back to me that top 0.05%, 0.1%, 0.5%, and 1% of the roads are selected due to physical constraints, and that is convincing in the real world. However, this information is not reflected in the latest paper version. I am focusing on this due to the reader gap that the audience may feel the authors hide something important by selecting the top 1% of the road only, how about the rest 99% (missing due to bad model performance or other reasons)? It is important to introduce the physical condition to the audience (WITH REFERENCES / supporting materials if possible). The same applies to other points.
(2) Figure 4 ABC, I would recommend removing the first column of all three sub-figures, as these indeed can not tell the audience much. Please remove outliers on the rest of the sub-figures as well (these are small in terms of percentage, while huge in terms of total count, and may lead to some misunderstanding).
(3) I have also reviewed your supplementary document, thank you for making these as a response. I would recommend to include these in your appendix section. Please make sure that figures are revised following comment (2)
Summary - thanks for the feedback and revisions, while those challenge points of the first version need to be fixed in your paper with reference support.
Author Response
Comment 1:
I appreciate you getting back to me that top 0.05%, 0.1%, 0.5%, and 1% of the roads are selected due to physical constraints, and that is convincing in the real world. However, this information is not reflected in the latest paper version... It is important to introduce the physical condition to the audience (with references/supporting materials if possible). The same applies to other points.
Response:
We appreciate the reviewer’s thoughtful follow-up on this important point. In the revised manuscript, we have now added explicit discussion on the practical constraints that motivate our choice of using only the top 0.05%–1% of roads ranked by betweenness centrality. Specifically, we noted that real-world urban sensing systems typically cover only a small portion of the road network due to budgetary and technical limitations. For example, large-scale deployments of loop detectors, GPS-based probes, or intelligent transportation infrastructure often monitor less than 1% of road segments in many cities (see e.g., Zheng et al., 2016, Fei et al., 2007). We have added this justification to Section 3.5 along with a citation and a brief note on sensor coverage. This clarification helps readers understand that the selection of top roads is rooted in physical and managerial constraints rather than arbitrary model tuning or exclusion of data. We have also reviewed the other points raised in our previous response and ensured that important clarifications are now reflected in the main text as well.
Comment 2:
Figure 4 ABC, I would recommend removing the first column of all three sub-figures, as these indeed can not tell the audience much. Please remove outliers on the rest of the sub-figures as well (these are small in terms of percentage, while huge in terms of total count, and may lead to some misunderstanding).
Response:
We thank the reviewer for the helpful suggestions. As recommended, we have revised Figure 4 by removing the first column in all three sub-figures (A, B, and C) to focus on the most informative categories. Additionally, we have also removed all the first columns of relevant sub-figures in Appendix. The figure caption and corresponding discussion in Section 3.4 have been updated accordingly.
Comment 3:
I have also reviewed your supplementary document, thank you for making these as a response. I would recommend to include these in your appendix section. Please make sure that figures are revised following comment (2).
Response:
We thank the reviewer for the suggestion. We have now moved the relevant figures and tables from the supplementary materials into a new Appendix section at the end of the main manuscript. This ensures better accessibility and alignment with the core narrative. We have also revised all figures included in the appendix to be consistent with the changes requested in Comment (2), including the removal of uninformative columns and outliers.